# Individualized Multimodal Immunotherapy (IMI): Scientific Rationale and Clinical Experience from a Single Institution

**DOI:** 10.3390/biomedicines12040754

**Published:** 2024-03-28

**Authors:** Volker Schirrmacher, Stefaan Van Gool, Wilfried Stuecker

**Affiliations:** Immune-Oncological Center Cologne (IOZK), D-50674 Cologne, Germany; vangool@iozk.de (S.V.G.); stuecker@iozk.de (W.S.)

**Keywords:** antigen-presenting cell, bone marrow, oncolytic virus, dendritic cell vaccine, electrohyperthermia, immunogenic cell death, glioblastoma, cancer-immunity cycle, memory T cell

## Abstract

Oncolytic viruses and combinatorial immunotherapy for cancer (this Special Issue) are both part of cancer treatment at IOZK. This review focusses on an individual multimodal cancer immunotherapy concept developed by IOZK, Cologne, Germany. The scientific rationale for employing three main components is explained: (i) oncolytic Newcastle disease virus, (ii) modulated electrohyperthermia and (iii) individual tumor antigen and oncolytic virus modified dendritic cell vaccine (IO-VAC^R^). The strategy involves repeated cancer-immunity cycles evoked in cancer patients by systemic oncolytic virus exposure plus hyperthermia pretreatment to induce immunogenic cell death followed by intradermal IO-VAC^R^ vaccination. As an example of the experience at IOZK, we present the latest results from combining the immunotherapy with standard treatment of patients suffering from glioblastoma multiforme. The promising clinical results in terms of overall survival benefit of additional individualized multimodal immunotherapy are presented. The cancer-immunity cycle, as introduced 10 years ago, describes key important steps occurring locally at the sites of both tumor and draining lymph nodes. This view is extended here towards systemic events occuring in blood where immunogenic cell death-induced tumor antigens are transported into the bone marrow. For 20 years it has been known that bone marrow is an antigen-responsive organ in which dendritic cells present tumor antigens to T cells leading to immunological synapse formation, tumor antigen-specific T cell activation and memory T cell formation. Bone marrow is known to be the most prominent source of de novo cellular generation in the body and to play an important role for the storage and maintenance of immunological memory. Its systemic activation is recommended to augment cancer-immunity cycles.

## 1. Introduction

The clinical success of immune checkpoint inhibitory antibodies has established immunotherapy as a new type of cancer therapy. This success, however, is incomplete. As such, other therapies, such as oncolytic viruses and combinatorial approaches, are being further investigated.

Cancer represents a group of neoplastic diseases derived from distinct cells and tissues of the body. Several hallmarks of cancer have been described [1]. The following have significance for the present review: (i) Sustained proliferative signaling, (ii) resisting cell death, (iii) activating invasion and metastasis, and (iv) avoiding immune destruction. With regard to the latter point, a recent review [2] has focused on T cells as key effectors of anti-cancer immunity. It describes the molecular mechanisms by which cancer cells evade T cell mediated immune destruction by gaining control over pathways that usually serve to maintain physiological tolerance to the self. This includes control over T cell localization, antigen recognition and acquisition of optimal effector function [2] (see Section 4.1).

Coping with the complexity of cancer requires combinatorial approaches of treatment, such as standard treatments like surgery, radio- and chemotherapy plus novel types of immunotherapy. Oncolytic viruses (OVs) and cancer vaccines are promising immunotherapeutics. They have been reported to exert profoundly lower side effects in cancer patients than other systemic therapies [3]. Side effects of chemotherapeutics and targeted therapeutics such as small molecule inhibitors often affect the immune system, in particular the important central immune system bone marrow (BM) [4]. They also cause therapy resistances because cancer is heterogenous and capable of developing cell-intrinsic mechanisms to resist cell death and to escape single targeted therapeutics and single targeted immune therapeutics [2]. Thus, resistance to therapy is a major obstacle to effective cancer treatment. The oncolytic Newcastle disease virus (NDV) has been described to be capable of breaking therapy resistance [3].

This review focuses on the combinatorial immunotherapy approach developed by IOZK. Therefore, those pioneering works receive particular detail and attention.

## 2. Individualized Multimodal Immunotherapy (IMI) Strategy

The new strategy of individualized multimodal cancer immunotherapy (IMI) was introduced in 2017 [5]. It combines hyperthermia/oncolytic virus pretreatment with specific autologous anti-tumor vaccination. The oncolytic virus employed, NDV, combines anti-neoplastic with immune stimulatory properties.

### 2.1. Newcastle Disease Virus (NDV)

NDV is an avian paramyxovirus type 1. Its genome is a negative sense single-stranded RNA. It is used for prophylactic vaccination in poultry and as oncolytic virus and adjuvant for therapeutic vaccination in cancer patients [3]. Normal healthy cells from birds are permissive for viral replication while those from mammals (e.g., mouse and human) are non-permissive. Over 50 years of clinical application of oncolytic NDV attest its low side effects and its high safety profile [3]. Clinical applications included single case observations, case series studies and Phase I to III studies [3]. Lack of interaction with host cell DNA, lack of genetic recombination and independence of virus replication from cell proliferation are viral characteristics supporting the safety profile [3].

#### 2.1.1. Tumor-Selective Virus Replication and Oncolysis

Some NDV strains show in tumor cells from cancer patients an oncotropism with tumor-selective virus replication and oncolysis. Molecular mechanisms of the tumor-selective virus replication of NDV have recently been reviewed [6]. They include (i) the endocytic targeting of the GTPase Rac1 in Ras-transformed human tumorigenic cells, (ii) a switch from cellular protein to viral protein synthesis and the induction of autophagy, mediated by the viral nucleoprotein NP, (iii) virus replication mediated by the viral RNA polymerase (L) associated with phosphoprotein (P), (iv) facilitation of NDV spread in tumors via membrane-budding of virus progeny with the help of matrix protein (M) and fusion protein (F), (v) oncolysis via apoptosis, necroptosis, pyroptosis or ferroptosis associated with immunogenic cell death (ICD) [6].

#### 2.1.2. Innate Immunity Stimulation

In contrast to tumor cells, many of which are sensitive to NDV replication, normal healthy non-tumor cells are resistant to NDV replication. This is due primarily to the induction of a strong type I interferon (IFN-I) response and the induction in the cells of an anti-viral state [6]. Recognition of viral RNA in the cytoplasm by RIG-I initiates the response via induction of IFN-I [6]. Recognition of secreted IFN-I at the cell surface by type I IFN-I receptor then initiates a cellular response amplification loop [6]. Signaling through RIG-I and type I IFN-I receptor causes immune activation by NDV in humans. Ebola virus inhibits the same signaling pathways to achieve immune evasion [7].

Interestingly, dysregulated cancer cells usually produce less IFN-I than normal cells upon contact and infection by viruses such as NDV. This phenomenon can be used by oncolytic NDV to cross the interferon barrier in tumor cells and to target and destroy a broad spectrum of different tumor cells from solid tumors [8]. 

Cellular and molecular mechanisms involved in NDV-mediated immune stimulatory activity have recently been reviewed [6]. They can be summarized as follows: (i) Increase in adhesive interactions between infected tumor cells and immune cells, (ii) Activation of NK cells by specific binding of HN to NKp44/46 receptors followed by signalling, (iii) Activation of monocytes and macrophages via NFkB mediated upregulation of tumor necrosis factor-related apoptosis inducing ligand (TRAIL) and secretion of tumor necrosis factor (TNF) alpha and nitric oxide (NO), (iv) Reprogramming of dendritic cells (DCs) to DC1 by a choreographed cascade of transcription factors and induction of a module for antigen presentation [9], (v) Viral oncolysate (VOL) uptake by DCs and promotion of antigen cross-presentation [6], and (vi) oncolysis-independent immune stimulatory effects with pro-inflammatory and abscopal effects [10].

#### 2.1.3. Adaptive Immunity Stimulation

The effects of NDV on anti-tumor responses of T cells are also multiple and have been reviewed [11]. They can be summarized as follows: (i) Augmentation of tumor cell immunogenicity, (ii) Augmentation of CD8+ T cell costimulation, (iii) Breakage of CD4+ T cell tolerance to tumor antigens (TAs), (iv) Augmentation of cooperative interactions between CD4+ T helper and CD8+ cytolytic T cell precursors (CTLPs), (v) Increase in frequencies of CD4+ and CD8+ CTLPs via induction of IFN α/ß, (vi) Cognate interaction of DC1 TA-presenting cells with CD4+ T cells leading to TA-specific activation and Th1 polarization, (vii) Th1 CD4+ T cells interacting with CD8+ T cells helping their differentiation into cytotoxic T cells (CTLs) and CD8+ memory T cells, and (viii) Induction of long-term protective anti-tumor immunity [11].

#### 2.1.4. Repeated Cancer-Immunity Cycles

The mechanism of the anti-tumor activity of oncolytic NDV has been exemplified with a migratory and invasive glioblastoma multiforme (GBM) tumor cell [6]. The direction of GBM cell movement is accompanied by an increase in Rac1 expression from the trailing edge to the leading edge at the lamellipodia. The first step consists of macropinocytosis/endocytosis of NDV targeting the small Rho GTPase Rac1. In the second step, the cap-dependent translational machinery is targeted through the MNK1/2-eIF4E axis by viral mRNA. This leads to viral protein translation in the cytosol and later in the double-membraned autophagosome. The third step consists of tumor-selective virus replication in autophagosomes. Intact viral particles are then produced in the fourth step. This involves virus progeny encapsulation, budding and virus release from the plasma membrane mediated via M, HN and F. In step 5, contact between released NDV and healthy normal cells leads to IFN-I secretion and to DC1 and Th1 polarization of adaptive immunity responses. The sixth and last step consists in NDV-induced tumor cell death (oncolysis) responses. These involve extrinsic (immune-mediated) and intrinsic cell death signaling pathways. ICD-derived components feed into antigen-presenting cells (APCs) that present TAs to T cells. Several rounds of such cycles (1–6) drive oncolytic effects and lead to immunological memory and systemic antitumor immunity.

#### 2.1.5. Breaking Cancer Therapy Resistances

Resistance to therapy is a major obstacle to cancer treatment. The potential of NDV to break therapy resistance has been reported [3]. It can be summarized as follows:Non-immune related resistances: (i) Chemotherapy and radiotherapy mediated resistances. These therapies target proliferating cells. Oncolysis by NDV does not depend on cell proliferation since this RNA virus replicates in the cytoplasm. There is thus a potential for oncolytic NDV to target cancer stem cells and dormant tumor cells. (ii) Apoptosis resistance. NDV was demonstrated to be capable to break resistance to apoptosis. (iii) Hypoxia resistance. Solid tumor microenvironments (TME) contain regions of hypoxia, in which a transcription factor, hypoxia inducible factor (HIF), is active. It influences gene expression and contributes to the tumor’s radio- and chemo-resistance. By testing a renal cell carcinoma line under normoxic and hypoxic conditions, it was found that hypoxia augmented oncolytic activity regardless of the cells HIF levels [12].Immune related resistances: (i) TRAIL resistance. TRAIL-resistant hepatocellular carcinoma-derived cell lines were found to be more susceptible to NDV-mediated oncolysis than TRAIL-sensitive cells. (ii) Immune checkpoint blockade (ICB) resistance. Intratumoral application of NDV in B16 mouse melanoma could break systemic tumor resistance to ICB [10]. The therapeutic effect was associated with marked distant tumor infiltration with activated CD4+ and CD8+ effector but not regulatory T cells. It was dependent on NK cells, CD8+ T cells and IFN-I. (iii) Anti-viral immunity resistance. Anti-viral immunity is considered as a major hurdle for effective therapeutic activity of OVs. Surprisingly, pre-existing immunity to oncolytic NDV was recently reported to potentiate rather than to inhibit its immunotherapeutic efficacy [13].

As stated in the conclusion of Section 2.1.1, Section 2.1.2, Section 2.1.3, Section 2.1.4 and Section 2.1.5, oncolytic NDV is an interesting agent to be included into a multimodal approach of cancer immunotherapy. It has a broad spectrum of anti-neoplastic activities. In addition, it has immunostimulatory and immunomodulatory potential. Our rationale [5] is to first use oncolytic NDV systemically (intravenously, i.v., bolus injection) for immunomodulation (induction of IFN-I, which inhibits secretion of Th2 cytokines (IL-4, IL-5) and facilitates Th1 polarization) and conditioning of the cancer patient’s immune system before employing a DC vaccine for active-specific vaccination. Conditioning includes targeting cancer cells in metastasized tissues and induction of immunogenic cell death (ICD) (see Section 2.3).

### 2.2. Modulated Electrohyperthermia (mEHT)

Since the efficiency of targeting cancer cells in metastasized tissues after intravenous virus application is usually low, we decided to combine it with another modality to increase tissue effects. This other modality is modulated electrohyperthermia (mEHT). 

Electromagnetic fields induce thermal and non-thermal effects on cancer cells, forcing the antitumor effects of heat-shock proteins (HSPs) [14]. Local mEHT is performed at 13.56 MHz with a power of 40 to 100 Watt for 20 to 60 min. Both i.v. bolus injections with NDV (1 to 10 × 10^7^ infectious particles) and sessions of mEHT (40 to 60 Watt for 50 min) are applied together as ICD immunotherapy, for patients with GBM [15].

mEHT can enhance virus tumor targeting [16] and virus replication [17]. It also mobilizes CTL migration and effector function in the TME [18].

In conclusion, mEHT has the potential to improve the targeting of NDV to the tumor-inflicted tissue. It also has the potential to improve ICD in tumor cells and the activity of CTLs.

### 2.3. Immunogenic Cell Death (ICD) Immunotherapy

ICD describes a specific type of Regulated Cell Death. It causes an adaptive immune response specific for endogenous (cellular) or exogenous (e.g., viral) antigens expressed by the dying cell [19]. It can be induced by virus infection, chemotherapeutics like anthracyclines, electromagnetic waves (mEHT) and others. The mechanism of ICD is based on timely controled release of so-called Damage Associated Molecular Patterns (DAMPs), which can be recognized by respective Pattern Recognition Receptors (PRRs). Those expressed by innate and adaptive immunity cells warns the organism of a situation of danger and elicit an immune response generally associated with the establishment of immunological memory. 

Six DAMPs appear to play a crucial role for ICD: (i) expression of calreticulin on the membrane, an “eat me” signal for phagocytosis by macrophages, neutrophils and DC precursors, (ii) secretion of ATP, a “find me” signal for macrophages and DC precursors, (iii) secretion of high mobility group box 1 (HMGB1) protein, which binds mainly to TLR4, an “approach me” signal, (iv) secretion of broadly immune stimulatory IFN-I, (v) release of cancer cell-derived nucleic acids, which are taken up by DCs, macrophages and neutrophils, and (vi) expression of annexin A1, a “recognize me” signal for DCs expressing formyl peptide receptor 1 [15].

Upon tumor cell infection by oncolytic NDV, virus-derived PAMPs contribute to ICD. For instance, a distinct viral RNA, ppp-RNA Leader of NDV, activates retinoic acid inducible gene I (RIG-I) to induce IFN-I responses [6]. The membrane expressed protein HN is recognized by the receptor NKp46 and activates NK cells [11]. In general, virus infection and exposure to PAMPs and DAMPs result in the activation of interferon-regulatory factor 3 (IRF3), which causes the induction of IFN-I and IFN-stimulated genes (ISGs). In non-tumorigenic cells, ISGs such as PKR, RNase L, ISG15, TRIM19 and TDRD7 can inhibit viral replication in different ways, including their ability to inhibit autophagy [20].

### 2.4. From First to Second Generation NDV-Modified Vaccine

The importance of individual tumor antigens (e.g., tumor rejection antigens, tumor neoantigens) and of the immunogenicity increase of tumor cells by infection with NDV was discovered in the 1980s in animal tumor studies [21]. This scientific knowledge was thereafter translated to clinical application with the autolopous NDV-modified vaccine ATV-NDV. This first-generation vaccine, consisting of irradiated NDV infected tumor cells, was used for post-operative immunization. Promising results were obtained in various clinical studies including a randomized-controled study [22].

The IMI strategy applied at IOZK makes use of an NDV-modified TA loaded DC vaccine as a second generation NDV cancer vaccine [5]. The advantage is the use of professional antigen-presenting cells (APCs) instead of tumor cells. The DCs are loaded with viral oncolysate (VOL).

### 2.5. ICD Products from Blood as Source of Individual Tumor Antigens (TAs)

A prerequisite for both types of vaccine (ATV-NDV and VOL-DC) was the availability of tumor material from operation specimens. Meanwhile, a third procedure is being developed at IOZK. The source of autologous TAs to load DCs now comes from blood (plasma) samples of pretreated cancer patients. ICD products are generated within 5-day treatment cycles of systemic NDV plus mEHT [15]. These repeated cycles of ICD treatment lead to the accumulation of extracellular microvessels (EVs) and apoptotic bodies in blood plasma. Such EVs potentially contain TA/MHC molecules on membrane fragments as well as DAMPs like HMGB1, HSPs and S100 proteins [15].

### 2.6. IO-Vac^R^, Individual NDV Modified Dendritic Cell Vaccine

For vaccine production, blood derived monocytes are purified via adherence, and are differentiated toward immature DCs in the presence of IL-4 and GM-CSF. Immature DCs are then loaded with Tas derived either from VOL or from ICD-induced plasma-derived Evs and apoptotic bodies. Finally, DCs are matured in the presence of IL1ß, TNFα, IL-6 and NDV. This autologous cell product is the vaccine IO-Vac^R^, which is administered to the patient intradermally [15,23,24,25]. 

IOZK received formal approval to produce this individual patient-specific vaccine as Advanced Therapeutic Medicinal Product (ATMP) for use in humans on 27 May 2015. This included the first production of NDV by Good Manufacturing Practice (GMP).

### 2.7. Bone Marrow as a Priming Site for T-Cell Responses to Blood-Borne Antigen

This is the title of a Nature Medicine paper from 2003 [26]. This had an impact on the IOZK strategy. The in-situ ICD-generated EVs and apoptotic bodies are not only useful to load the vaccine IO-VAC^R^. They also immediately circulate via blood to the BM [4]. There they have the potential to prime cancer-reactive naïve T cells as well as to reactivate pre-existing cancer-reactive memory T cells (MTCs) that reside in distinct niches of the BM. Twenty years after the original article, new discoveries about the BM have been summarized in a recent review, entitled: Bone marrow: The central immune system (CIS) [4].

## 3. Towards a Multiphase Combined Treatment Strategy for GBM Patients

### 3.1. Glioblastoma Multiforme (GBM)

For GBM, an orphan disease with poor outcome, an immune landscape has been described as a double-edged sword for treatment [27]. On one hand, there are the immune-stimulatory effects of lymphocyte responses against glioma cells; on the other hand, there are the immunosuppressive effects of tumor cells, myeloid suppressor cells and others [27].

### 3.2. Individualized Multimodal Immunotherapy (IMI) as Part of First-Line Multiphase Combined Treatment for GBM 

A rational novel combined treatment strategy for GBM was designed [6,20] based on the standard of care (neurosurgery, chemotherapy and maintenance temozolomide (TMZm) therapy) and the above described components NDV, mEHT and IO-Vac**^R^_._**

1. The first phase is directed towards optimization of anti-cancer activity beyond monotherapy with alkylating agents such as temozolomide (TMZ). It includes bolus injection of NDV and sessions of mEHT to induce ICD.

2. The second phase starts after chemotherapy is finished; this is the immunization phase with the vaccine IO-Vac**^R^**. This provides the antigenic spectrum of in vivo existing tumor cells, which persist despite radiochemotherapy and TMZm. Recently, this antigenic spectrum has been expanded with long-peptide vaccines covering some more generally present tumor antigens (e.g., WT1, survivin) [24]. Modulatory immunotherapy is implemented, depending on the situation, with curcumin, celecoxib and anti-histamin receptor-1 blockers.

3. The third phase is directed towards maintenance of the anticancer immune control, and to expand the covered antigenic spectrum. Repetitive 5-day ICD immunotherapy courses keep targeting and killing new developments of tumor cell clones. Since 2021, a boost vaccine, at least six months after the second IO-Vac**^R^**, is recommended to increase a memory response to TAs. During this phase, the modulatory strategies continue [15].

The IMI strategy includes several cancer-immunity cycles (see Section 4). Intradermal vaccination initiates local events around the vaccination site and its draining lymph nodes. The ICD treatment with NDV and mEHT stimulates systemic events causing accumulation of EVs and apoptotic bodies in blood plasma. These circulate to BM where cancer-reactive T cell responses are primed and boosted. Several such cycles lead to the accumulation of cancer-reactive memory T cells (MTCs), which are maintained in special niches of the BM [4]. They are the basis for long-term systemic anti-cancer immunity [11]. 

### 3.3. Clinical Results

In 2023, we were able to retrospectively analyze data from a group of 50 adults with isocitrate dehydrogenase 1 (IDH1) wild-type GBM [25]. Information about O^6^-methylguanine-DNA-methyltransferase (MGMT) promoter methylation status and follow-up information on overall survival was available. All consecutively treated patients were evaluated, without any selection. The data reflect real-world data (RWD) [23]. 

The median overall survival (mOS) of this first study was 27 months. The 2-year overall survival (OS) was 57.9% and the 3-year OS was 37.1%. 

These OS results were discussed in the context of external control arms of contemporary randomized-controlled trials (RCTs) as state-of-the-art in 2024 [28]. Data are shown in Table 1. Standard data from 2024 are improved in comparison to those from 2009 [29]. With the implementation of IMI integrated during maintenance chemotherapy (CT) and continued after standard of care (radiochemotherapy, RCT), we demonstrate in study I an increase in 2-year OS by a factor of 1.6 (unmethylated) and 2.7 (methylated) [15,25].

In Table 1, we present results from the latest analysis (study II) of this group of GBM patients, now expanded by 21 patients and followed for a longer period. Thirty-one (15 female, 16 male) patients were MGMT-promoter methylated, while 40 (14 female, 26 male) were MGMT-promoter unmethylated. Median age for both groups were not different: 54 years (range 26–72) and 48 years (range 18–65). The Karnofsky performance index in both groups was similar: 80 (range 60–100) and 70 (range 50–100). The distribution of patients qualified as less than completely resected, completely resected, not documented were equal in both groups: 17/11/3 for the MGMT-methylated patients versus 17/15/8 for the MGMT-unmethylated patients. The time between diagnosis at neurosurgery and uptake for IMI was equal for both groups: in median 3.5 months for MGMT-promoter methylated patients and 3.4 months for MGMT-promoter unmethylated patients. Most patients entered the additional IMI treatment after the radiochemotherapy at start of the first or second TMZ maintenance cycle. The OS in this cohort of RWD patients was calculated versus time of first diagnosis.

In all four columns a clear positive effect of IMI versus standard can be seen. We demonstrate an increase in 2-year OS by a factor of 1.5 (unmethylated) and 2.9 (methylated). While IMI has a positive effect on both subgroups, the MGMT methylated group appears to profit more from it than the MGMT unmethylated group.

It can be concluded from the results obtained at IOZK and published continuously from 2017 to 2023 [6,15,24,25] that the standard of care of GBM is strengthened and improved with IMI. The concept is therefore continued. Whenever possible, new discoveries can be integrated for further optimization.

### 3.4. Distinct Profile of IMI Treatment at IOZK

-All patients at IOZK are treated on an individual basis.-This is possible because there exists a special legal framework for this in Germany.-All consecutively treated patients are being evaluated, without any selection. The obtained data represent real-world data (RWD).-The IMI strategy of IOZK includes the use of GMP produced oncolytic avian virus NDV.-The side effects of IMI are very low (grade 0–2)-A therapeutic effect of IMI is demonstrated by combining first-line standard treatment with IMI in adults with GBM.-Results of IMI have also been reported from single case studies at IOZK of other cancers: (i) long-term remission of a prostate cancer patient with extensive bone metastases [5], (ii) long-term survival of a breast cancer patient with extensive liver metastases [5].

### 3.5. Real-World Data and Real-World Evidence

Evidence-based medicine is often solely based on so-called “best research evidence”, collected through RCT while disregarding clinical expertise and patient expectations. Such external clinical evidence can inform, but not replace, individual clinical expertise. This applies in particular to orphan diseases like GBM, for which clinical trials are methodologically problematic [23]. In the fast-changing therapeutic landscape and the emergence of immuno-oncology therapies for numerous cancer types, there is a need to rapidly assess new agents through real-world evidence [30].

### 3.6. Overview of Different Types of Cancer Treated with NDV-Modified Cancer Vaccines

The DC vaccine IO-VAC represents a second-generation vaccine. The first-generation vaccine was the autologous NDV-modified tumor cell vaccine ATV-NDV [21,22], which was developed and studied by the first authors group at the German Cancer Research Center in Heidelberg, Germany. Table 2 shows the outcome of several phase II studies that have been performed with different types of solid human cancers. 

With the exception of rectum carcinoma, all studies revealed improved OS and disease-free survival (DSF) in comparison to historical or concomitant controls. At the time, the concepts of autologous versus heterologous (allogeneic) cancer vaccines were heavily disputed. Because autologous vaccines are more cumbersome to prepare, it was felt to be unlikely that this approach might be successful. The IMI strategy at IOZK is applied to about 70 different types of human cancer, mostly solid tumors. Most patients come from Europe. From the n = 3329 records of patients in 2024, 23% are neurological cancers, followed by digestive tract cancers (22%), breast cancers (17%) and urological cancers (12%). It will take time to evaluate the clinical outcome of cancers other than GBM. Single case responses have been observed and some have been published [5].

### 3.7. Oncolytic Virus Clinical Trials

By the end of 2023, nearly 180 clinical trials of oncolytic virus (OV) therapy have been registered. Adenovirus, vaccinia virus and oncolytic type 2 Herpes simplex virus are currently in Phase III and IV trials. The five most common viruses registered between 2000 and 2023 are adenovirus, HSV-1, reovirus, vaccinia virus and NDV [37]. There are natural OVs such as NDV-HUJ, Reovirus (Reolysin) and Protoparvovirus H-1PV, which are injected either intravenously (NDV) or intratumorally. Then there are recombinant OVs such as Poliovirus (PVS-RIPO), HSV (HSV1716, G207), Adenovirus (DNX-2401) and Measlesvirus (MV-CEA). Finally, there are examples of OVs in early development such as Vacciniavirus (TG6002), Coxsackievirus (CVB5) and Seneca Valley Virus (SVV-001). 

## 4. The Cancer-Immunity Cycle

We like to reflect the IMI strategy in the light of recent new immunotherapy concepts such as the cancer-immunity cycle [38].

### 4.1. Local Events: Tumor and Draining Lymph Nodes

With regard to local events in the TME, an excellent recent review describes the molecular details of cancer cell-intrinsic mechanisms driving acquired immune tolerance [2]. Nine mechanisms are distinguished: (i) Stromal inhibition of T cell recruitment, (ii) reduced production of chemokines involved in T cell recruitment, (iii) inhibition of target recognition: suppression of MHC class I expression, (iv) immune escape via suppression of neoantigen (TA) expression, (v) impaired target recognition through suppression of DC recruitment, (vi) limiting the attainment of optimal T cell effector function, (vii) antigen diversity as a driver of T cell dysfunction, (viii) effector diversion and recruitment of suppressive populations, and (ix) direct induction of T cell death and co-inhibitory signaling. It is further concluded that cancer cell evolution converges on immune evasion strategies that either replicate or mimic pathways of peripheral tolerance [2]. The cancer-immunity cycle [38] is capable of overcoming and breaking the above acquired immune tolerance mechanisms. Its seven steps are the following: (i) Release of cancer cell antigens (cancer cell death), (ii) cancer antigen presentation (dendritic cells/APCs), (iii) priming and activation (APCs and T cells), (iv) trafficking of T cells to tumors (CTLs), (v) infiltration of T cells into tumors and stroma, (vi) recognition of cancer cells by T cells, and (vii) killing of cancer cells (immune and cancer cells) [6,38]. Meanwhile, a TME cancer-immunity subcycle has been added (tertiary lymph node structures, or TLS). These distinguish inflamed from immune deserted or immune excluded tumor tissue [38]. 

### 4.2. IMI and the Cancer-Immunity Cycle

The seven steps of the cancer-immunity cycle appear to be fulfilled by the IMI strategy. The ICD therapy leads to the release of cancer cell antigens (TAs, EVs) into the blood (step (i)). From there they are transported into BM, where steps (ii) and (iii) occur.

Steps (iv) to (vii) also appear to be fulfilled. In 2004, Steiner et al. [36] published the earliest report on the results of antitumor vaccination of GBM patients with NDV modified vaccine (ATV-NDV). Immune monitoring of vaccinated and non-vaccinated patients revealed (i) a significant six-fold increase in infiltration of CD8+ T cells in relapsed tumors of vaccinated patients in comparison to non-vaccinated patients (steps iv and v), (ii) a significant augmentation of the frequencies in peripheral blood of tumor-reactive memory T cells (IFN-gamma response, ELISPOT) (step vi) and (iii) tumor regression as evidenced by magnetic resonance imaging: a tumor that developed in a patient during radiotherapy had completely disappeared six months after vaccination (step vii) [36].

The presence of TLS as an amplification loop will be discussed below with respect to events in the BM (see Section 5.2).

### 4.3. Breaking Acquired T Cell Tolerance by IMI

How immunosuppression in the TME can be counteracted by oncolytic NDV and IMI has been reviewed [39]. Twelve anti-neoplastic effects of NDV in non-permissive hosts as well as 11 immune stimulatory effects have been described [3]. Of importance, with regard to the above review [2], is the fact that oncolytic NDV can break T cell tolerance to TA-expressing tumor cells and to break resistance to immune checkpoint blockade. 

In conclusion to the reflection about IMI and cancer-immunity cycles, it can be stated that IMI contains several cancer-immunity cycles: at the site of the tumor cell after systemic NDV application, at the lymph node draining the vaccination site and in the bone marrow (see Section 5).

## 5. Systemic Events: Blood and Bone Marrow

While most of the recent reviews from “Immunity” (6 October 2023) about the cancer-immunity cycle focus on events in tumor and tumor-draining lymph nodes, hardly any attention is directed towards blood-borne antigens and how the immune system deals with these. Since IMI includes systemic immune stimulation, it is likely to stimulate immune responses from the BM. It is thus important to have a closer look into such systemic events.

### 5.1. Bone Marrow: The Central Immune System

Extracellular fluid (lymph) is constantly drained from peripheral tissues through lymphatic vessels into lymph nodes. Thereafter it reaches the bloodstream via the thoracic duct; 75% of the lymph from the entire body is transported through this largest lymphatic vessel [4]. Self-antigens (SAs) transported through blood to the BM and thymus are involved in negative selection of potentially self-reactive B or T cells [4].

That BM is not only a hematopoietic but also an antigen-responsive lymphatic organ was discovered already 20 years ago [26]. Among all antigen-responsive immune organs BM is the largest, comprising 4–5% of the total body weight [4]. BM is the most prominent source of de novo cellular generation, reaching rates of 4–5 × 10^11^ cells per day in an adult human [4]. 

Blood-borne antigens include: (i) peripheral and systemic antigens, (ii) immune complexes, (iii) macromolecules, and (iv) tumor-associated proteins. Blood circulatory cells homing to BM include: (i) cells infected by blood-borne viruses (e.g., HCV, HBV, HIV), (ii) circulatory tumor cells and derived EVs and apoptotic bodies, (iii) circulatory APCs, and (iv) naïve and memory T cells. 

BM from untreated breast cancer patients was reported to be enriched with memory T cells [4]. Breast cancer induced cancer-reactive MTCs could be reactivated ex vivo and shown in 2001 to confer therapeutic activity upon adoptive transfer to NOD/SCID mice with xenotransplanted human breast cancer [40]. These and associated studies demonstrated selective homing of human MTCs to human tumors in xenotransplanted mice. Tumor rejection was based on the recognition of TAs on tumor cells and DCs by autologous specifically activated central and effector MTCs [41].

The steps of T cell activation in BM parenchyma are similar to those of the cancer-immunity cycle, but distinct for the BM: (i) Homing of T cells and antigen to BM, (ii) DC presentation of self (SA) or non-self (NSA) antigens, (iii) APC scanning by T cells, (iv) T cell response (tolerance or activation), (v) T cell proliferation, and (vi) T cell recirculation [4].

It is now becoming clear that DCs play a role not only in lymph nodes and in the TME but also at other sites [42], in particular in the central immune organ BM [4]. Cognate T-APC interactions occur in perivascular sinusoidal parenchyma niches of BM upon immunological synapse formation, with centrosome polarization and signaling events via signaling complexes [4].

### 5.2. Bone Marrow and Tertiary Lymph Node-like Structures

TLS can be observed in BM parenchyma upon antigenic stimulation. BM clusters in the BM can develop into large follicles [40,41]. These include memory B and memory plasma cells in addition to CD4+ MTCs, suggesting T-B cell interaction. DCs and CD8+ T cells are also observed, suggesting APC-T cell interactions [4]. BM thus seems capable of establishing immune synapses and performing self-amplifying loops in follicles like TLS in the TME.

### 5.3. BM as an Autonomous Refuge for Immune Memory

BM-derived adaptive immune responses were found to be autonomous. Splenectomized mutant Map3k14aly/aly mice, which lack lymph nodes and payer’s patches, could activate CD8+ and CD4+ T cells in their BM upon antigen stimulation in the same way as mice with spleen and lymph nodes, respectively [26].

BM has been reported to function as a refuge for immune memory under dietary restriction (DR). MTCs collapsed in secondary lymphoid organs in the context of DR, but dramatically accumulated in the BM [43]. The response to DR included an increase in T cell homing factors, erythropoiesis, and adipogenesis. Homing of MTCs to BM during DR was associated with enhanced protection against infections and tumors [43]. 

BM is thus an autonomous immune organ. It is also a refuge for immune memory, the basis for long-term protective immunity.

Table 3 provides an overview of special features of the BM.

## 6. Conclusions

An individual multimodal cancer immunotherapy strategy, IMI, is presented to combat the cancer intrinsic immune escape and immune suppressive mechanisms. It consists of a combination of systemic and local immunization steps. The components are an oncolytic virus (NDV) with broad anti-neoplastic and immune stimulatory properties, modulated electrohyperthermia (mEHT) and a dendritic cell vaccine (IO-Vac^R^). Cycles of NDV plus mEHT treatment cause the accumulation of ICD products (extracellular microvesicles (EVs) and apoptotic bodies) in the patient’s blood plasma. These are transported to the BM where systemic anti-tumor immune responses are stimulated. ICD products are also harboured from plasma and combined with patient-derived DCs to produce a DC vaccine for local intradermal vaccination.

The strategy is based on several decades of basic and applied research at the German Cancer Research Center (DKFZ) Heidelberg. The first-generation vaccine ATV-NDV was developed there. The second-generation vaccine IO-VAC was developed at IOZK. It has been translated into clinical application since 2015, the year of legal approval in Germany. The data from the clinical results are published as single case studies or as real-world data from retrospective analyses. In the case of patients suffering from GBM, a three-phase treatment strategy is presented. The RWD data demonstrate a clear benefit of combining IMI with standard therapy.

IMI is superior to monotherapies because it includes an oncolytic virus with potential to break therapy resistances and because it is multimodal.

## 7. Take Home Messages

Box 1 summarizes the take home messages.

Box 1Take home messages.
Scientific rationale
(i)In the fight against cancer the immune system needs to be instructed and activated against the individual tumor antigens of the patient’s tumor. To achieve this, the IOZK strategy involves for instruction the induction of immunogenic cell death (ICD) and anti-tumor vaccination with tumor antigen-loaded dendritic cells (DCs) and for activation the use of oncolytic virus.(ii)Since cancers tend to change their immune phenotype over time, patients should be followed longitudinally by focussing the immune response against the actually present immune phenotype. This is possible by using ICD products from the plasma of treated patients to load the DCs.(iii)Individualized multimodal immunotherapy (IMI) generates so-called real-world data. These can be evaluated scientifically to generate medical evidence.
Clinical results
(i)One example is provided from patients suffering from glioblastoma multiforme (GBM), a rare but fatal disease.(ii)A retrospective analysis of unselected n = 71 IDH1 wildtype GBM patients treated with standard therapy plus IMI revealed a two-year overall survival (OS) of 42.7% for MGMT unmethylated and of 75.5% for MGMT methylated patients.(iii)These results can be compared to state-of-the art results from randomized-controlled studies with selected GBM patients treated by standard therapy without IMI. The two-year OS was 14.8% for MGMT unmethylated and 48.9 for MGMT methylated patients and thus clearly lower compared to the results obtained in combination with IMI.(iv)Another important point is that the increased therapeutic efficiency (factor 1.5 for unmethylated and factor 2.9 for methylated GBM) of combined IMI is associated with only grade 0–2 side effects.



## 8. Future Perspectives

The review describes an individualized multimodal immunotherapy, its scientific rationale in the context of latest immunotherapy concepts and clinical experience with it from a single institution in Germany. We like to draw attention to only two future aspects: (i) The use of EVs as source of TAs and (ii) the use of cancer-reactive MTCs from the patients BM.

One important innovative aspect that needs future development is the use of ICD-induced EVs from the patient’s plasma. The isolation, characterization and quantification of such EVs as a source of TAs needs to be established and validated. This would be useful for diagnostic and therapeutic applications. The production of a TA-loaded individualized vaccine would become independent from operated tumor material. The TAs of the vaccine would represent the relevant TAs at the time of patient vaccination.

In future, the IMI strategy could be extended by adoptive cell-mediated immunotherapy (ADI). We favor the use of cancer-reactive MTCs from BM because we have good experience from previous experimental work. In comparison to the use of peripheral blood cells, the use of cells from the BM for immunotherapy is a greatly neglected field in oncology. Cancer-reactive MTCs from BM were found to be superior to those from peripheral blood. Future clinical applications with focus on BM could be directed towards BM vaccination and BM MTC transplantation (autologous and/or allogeneic).

## Figures and Tables

**Table 1 biomedicines-12-00754-t001:** Overall survival of standard therapy for GBM with or without IMI.

	Study Type	Treatment	Unmeth. mOs (m)	Unmeth. 2y OS (%)	Meth. mOs (m)	Meth. 2y OS (%)	Ref.
Standard	RCT	S+RCT+CT	12.6	14.8	23.4	48.9	[28]
IOZK Study I (n = 50)	RWD	S+RCT+CT +IMI	22.1	39.4	37.7	80.5	[15,25]
IOZK Study II (n = 71)	RWD	S+RCT+CT +IMI	22.1	42.7	37.7	75.5	this paper

IMI = Individualized multimodal immunotherapy; m = months; mOS = median overall survival; RCT = randomized controlled trial; RWD = real world data; S+RCT+CT = surgery + radiochemotherapy + maintenance chemotherapy; 2y OS = 2-year overall survival (%); GBM = glioblastoma multiforme; Unmeth. = MGMT unmethylated; Meth. = MGMT methylated.

**Table 2 biomedicines-12-00754-t002:** Types of cancer treated post-operatively with the first-generation vaccine ATV-NDV.

Disease	Stage/Grade	Study Phase Patient Number	Clinical Outcome	Ref.
CRC	II and III	II (n = 57)	Improved OS and DFS	[31]
CRC	IV	II (n = 23)	Improved OS and DFS	[32]
Colon Ca	IV	II/III (n = 26)	Improved OS (*p* = 0.01)	[22]
Rectum Ca	IV	II/III (n = 25)	Not significant	[22]
Breast Ca	Locally advanced	II (n = 32)	Improved OS and DFS	[33]
Pancreas Ca	G3	II (n = 53)	Improved OS and DFS	[34]
HNSCC	III and IV	II (n = 18)	Improved OS and DFS	[35]
GBM	IV	II (n = 23)	Improved OS and DFS	[36]

All studies received approval by local ethical committees. The vaccines were applied intradermally. The vaccines (X-irradiated by 200 Gy) were applied multiple times without causing adverse events. The patient-derived (autologous) tumor cells were infected with the avirulent lentogenic NDV strain *Ulster*. CRC = Colorectal carcinoma; Ca = carcinoma; HNSCC = Head and Neck squamous cell carcinoma; OS = Overall survival; DFS = Disease free survival.

**Table 3 biomedicines-12-00754-t003:** The central immune system bone marrow.

Feature	Characteristics	Reference
Production of erythrocytes	Provision of oxygen and energy	Textbooks
Production of white blood cells	Provision of innate immunity cells and precursors of adaptive immunity cells	Textbooks
Homing of T cells and DCs to BM	Attachment to VCAM-1, ICAM-1 and selectins in BM microvessels	[26,40]
Blood-borne antigens	Self-antigens (SA), non-self antigens (NSA): Viral antigens (VA), tumor neoantigens (TA) ICD products, EVs, apoptotic bodies	[4]
BM antigen presentation by DCs	BM resident CD11c DC, and circulatory DC subset	[4]
BM T-APC interaction	Formation of immunological synapses; T-DC cluster formation	[4] [22,40,41]
BM immune responses	Antibody production, Effector T cells (Th, CTL)	[4]
BM storage of memory cells	Multiple niches created by stromal cells providing quiescence and survival signals	[4]
BM immune surveillance of the CNS	Antigen cross-presentation and T cell responses in calvaria, microglia network, SLYM	[44]
Adaptation to energy crisis	Recruitment of memory T cells from lymph nodes to BM	43

ICD = Immunogenic cell death; EV = Extracellular vesicle; DC = Dendritic cell; Th = CD4+ T helper cell; CTL = CD8+ Cytotoxic T cell; SLYM = Subarachnoid lymphatic-like membrane; BM = Bone marrow

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
