# Peer review of "Individualized Multimodal Immunotherapy (IMI): Scientific Rationale and Clinical Experience from a Single Institution"

_biomedicines, 2024, doi:10.3390/biomedicines12040754_

Round 1

Reviewer 1 Report

Comments and Suggestions for Authors

The author of this manuscript discusses the scientific rationale and clinical outcomes of cancer therapy in Germany using an individual multimodal cancer immunotherapy strategy. It is helpful to both scientists and clinical researchers. However, it would be easier for readers to understand if the author could revise the rationale section in a concise and logical manner. Aside from that, there are some issues to address.

1.    I recommend that the authors simplify the description of Table 1’s content in the text by focusing on the most important comparison data. Eliminating repeated descriptions of table 1 from the text will improve readability.

2.    Figure 1 does not need to be taken directly from another review article.

3.    The arrangement of subsections in Section 2 should be carefully considered. For example, in lines 175, 189, and 215, the section numbering appears incorrect.

4.    Box 1's take-home messages can be simplified by removing unnecessary sentences. For example, combining 1.i and 1.ii would increase brevity.

Comments on the Quality of English Language

There are some grammatical errors need to be corrected carefully.

Author Response

We thank all reviewers for their constructive criticism.

Response to reviewer 1:

  1. Table 1 and the accompanying text have been simplified.
  2. 1 has been eliminated.
  3. The section numbering of Section 2 has been corrected.
  4. Box 1`s take-home message has been simplified.

Reviewer 2 Report

Comments and Suggestions for Authors

The author's have prepared a detailed manuscript over the treatment of cancers through a multimodal approach with constant patient care. the review is rather detailed, and explains the process and pitfalls of the therapy involved. Since there is description from just one institute there is no comparative therapeutic approach compared to assess the level of success for the review of treatments. Hence it is rather difficult to assess on a stand alone basis the merit of the therapy. 

   Pitfalls: Since the data are from just one institution there should be tabulated data on different types of cancers that have been treated apart form the one mentioned(GBM), the success rate with that as well as patient outcomes if possible.

 In case there is no additional data, the title should be change to indicate that

3. Please compare other treatment modalities with oncolytic viruses vis-a-vis other viruses and state the importance of IO-VAC in therapy.

Author Response

We thank all reviewers for their constructive criticism.

Response to reviewer 2:

  1. A new paragraph (3.6.) and a new Table (Table 2)

have been added in the revised version. These show the different

types of cancer that have been treated with the first-generation

cancer vaccine ATV-NDV. Based on the positive findings, IOZK has

developed its second generation vaccine.

  1. A new paragraph (3.7.) has been added to provide a state-of-the-art

overview of current clinical developments with oncolytic viruses.

Reviewer 3 Report

Comments and Suggestions for Authors

Cancer has become a global burden, and yet several types of tumours did not have therapy. The individualized immunotherapy has become one of the possible therapies for cancer. Schirrmacher et al., have summarised the concept entitled “Individualized multimodal immunotherapy (IMI): Scientific rationale and clinical experience from a single institution”. The authors have summarized the multimodal immunotherapy results in the IOZK center, Germany. Despite tumour microenvironment and lymph node, the author's focus was on the system followed by bone marrow immunologic events. Overall, the manuscript is well written.

This reviewer has some suggestions.

Section 2.1: Please mention the characteristics/pathologies of co-infection probabilities of NDV with other viruses.

As discussed in Section 2.1.2, the innate immune system is high in recognizing the NDV. As a thumb a rule, if the virus is injected via a systemic route, the myeloid cells would capture and kill the virus before it reaches the tumour microenvironment, particularly in glioblastoma. How does the treatment strategy overcome this limitation?

Figure 1 is not an appropriate format for publication. In addition, the illustration could be improved with other cellular aspects, like autophagy or other cellular events to viruses.

Lines 134-137: This paragraph is contrary to the safety profile of the virus. If the virus or viral components are immunogenic, how it is safe? As each efficacious vaccine has some level of side effects. Please elaborate on the discussion on this aspect. In addition, please also discuss the immunogenicity of viral components comprehensively, e.g., nucleoprotein.

Section 2.5. second paragraph, are there any studies that reported on NDV infectivity in normal and tumour cells under hypoxia conditions?

The discussion should be supplemented with broad epidemiology data.

Author Response

We thank all reviewers for their constructive criticism.

Response to reviewer 3:

  1. A search for co-infection probabilities of NDV with other viruses in human

has failed.

  1. The limitations mentioned are correct. The treatment strategy includes mEHT

to increase tumor targeting efficiency after systemic NDV application. In future

there may be other means such as viral hitchhiking on carrier cells.

We include a new review (Ref. 25) which further discusses the issue of the tumor

microenvironment and how the immunosuppressive effects can be counteracted.

  1. 1 has been deleted.
  2. NDV has been selected among many other oncolytic viruses to increase the immunogenicity

of tumor cells, be it in a vaccine or in situ.  Immunogenicity increase of tumor cells has been a goal from the beginnings. As a bird virus, NDV has a high safety profile. It also has only low side effects. The HN molecules on infected tumor cells increase the adhesive interactions with immune cells. In contrast to human natural oncolytic viruses, natural NDV has no immunosuppressive effects in human cells. References 7, 9 and 13 are recommended for further details.

  1. NDV has been tested in clear cell renal carcinoma cells in normoxic and hypoxic conditions. Infection by NDV could break the hypoxia resistance. This is described in Ref. 18.
  2. It was not comprehensible to the authors why the discussion should be supplemented

with broad epidemiology data.

Round 2

Reviewer 1 Report

Comments and Suggestions for Authors

Despite the authors' attempts to address reviewers’ comments, the take home messages in the manuscript remains lengthy and lacks a clear focus on essential information. To enhance conciseness, I recommend removing the subsection i) In Clinical Results section. Additionally, consider shortening paragraphs throughout the text to emphasize essential information and improve overall readability.

Reviewer 3 Report

Comments and Suggestions for Authors

The authors have revised the manuscript adequately. Therefore, I endorse the manuscript for publication in its current form. 

Author Response

Letter to the reviewers, second round

Reviewer was satisfied with the revised version.